# Invasive Apple Snail Diets in Native vs. Non-Native Habitats Defined by SIAR (Stable Isotope Analysis in R)

**Kevin E. Scriber II [1,\*]**, **Christine A. M. France [2]** and **Fatimah L. C. Jackson [3]**

1   Department of Environmental Science, University of Arizona, (Shantz Building Rm. 429) 1177 East 4th St., Tucson, AZ 85721, USA
2   Smithsonian Museum Conservation Institute, 4210 Silver Hill Rd., Suitland, MD 20746, USA
3   Department of Biology, Howard University, (Ernest Everett Just Hall) 415 College St., NW, Washington, DC 20059, USA
\*   Correspondence: kevescriber@arizona.edu

**Abstract:** Invasive apple snails adversely impact the ecological function of non-native habitats, resulting in eutrophication as well as reduced biodiversity, which diminishes ecosystem goods and services, thereby [negatively] impacting human well-being. The onus here is to define the diet of an invasive apple snail (*Pomacea canaliculata*) in native (Maldonado, Uruguay) versus non-native habitats (Hangzhou, China, and Oahu, HI, USA). Diets for apple snails, in five sites, within both native and non-native habitats were defined via SIAR (Stable Isotope Analysis in R) with $\delta^{13}$C and $\delta^{15}$N stable isotope data collected therein. SIAR models indicate *P. canaliculata* shift diet from generalist (where myriad plant species comprise relatively small proportions of overall diet) to a specialist diet (where plants species constitute much larger proportions of said diet). What may be more telling is that in (anthropogenically disturbed) portions of the native habitat, and progressively more so in non-native habitats, invasive apple snail diets are increasingly composed of aquatic plants. The inherent and pronounced dietary differences amongst pristine and anthropogenically disturbed native habitats, as well as non-native habitats, provide a mechanism that may elucidate the variable ecological impacts of invasive apple snails within native and non-native habitats.

**Keywords:** invasive species; *Pomacea canaliculata*; aquatic ecology; trophic ecology; SIAR; aquatic biodiversity; aquatic ecosystems; freshwater food webs

## 1. Introduction

Numerous studies indicate the decline of ecosystem services and function has three major causes: (1) habitat destruction [1], (2) climate change [2,3], and (3) invasive species [4–6]. All cause precipitous declines in biodiversity [7]. As such, the onus to preserve ecosystem function and/or services should be placed on the systematic conservation of biodiversity and the focused restoration of the ecosystems that support it [8–10].

Ecosystem function and services are adversely impacted by invasive species [11,12], both in terms of the amenities they provide and the cost of their preservation and/or restoration [13,14], as coevolved trophic relationships and the natural flow of energy, materials, and nutrients throughout ecosystems are modified and/or destroyed. As such, invasive species are thought (by some) to be the leading cause of animal extinctions [15]; though data used to support this opinion are considered "anecdotal, speculative, and based upon limited observations" [16]. There are few clear connections between ecosystem services and invasive species [17,18]. This has produced an alternative view that habitat modification displaces native species and facilitates the influx of non-native species [19].

An invasive apple snail, *Pomacea canaliculata*, is listed amongst 100 of the world's worst invasive species and adversely impacts non-native freshwater habitats globally [20]. *P. canaliculata* consumes macrophytes in non-native habitats, diminishing ecosystem goods and services, including agriculture and the cultivation of staple crops such as rice, taro plant,

and watercress [21–24]. This invasive species causes eutrophication; reduces biodiversity, environmental heterogeneity, and human well-being [25–27]; and can adversely impact human health as a vector for disease (*Angiostrongylus cantonensis*; [28]). Generally, invasive apple snails (including *P. canaliculata*) do not cause adverse agricultural and/or ecological impacts in native habitats [29] (p. 215). However, *Pomacea dolioides*, which was previously misidentified as *P. lineata*, has impaired rice production in Suriname [30].

This study investigates why invasive apple snails (*Pomacea canaliculata*) differentially impact native and non-native habitats. Specifically, this work tests the hypothesis that *P. canaliculata* shift their trophic ecology and/or feeding behavior between native and non-native habitats. These observed shifts may facilitate a better understanding of the negative ecological impacts attributable to invasive apple snails. To that end, stable isotope data were essential to this study. The hypothesis being tested here is that the diet of *P. canaliculata* shifts from native to non-native habitats, from that of a generalist (where a multitude of available food resources in native habitats are utilized and represent smaller proportions of the overall diet's composition) to a more specialized diet (where fewer food resources are utilized and represent greater proportions of invasive apple snail diets in non-native habitats).

Heavy nitrogen isotopes are bioaccumulated, as they are incorporated into bodily tissues by consumers, while light isotopes are excreted in metabolic wastes (such as urea). This bioaccumulation is more pronounced in consumers placed higher in the food chains. Carbon isotopes are more indicative of the plant species that produce them, as carbon isotopic ratios vary according to plants' photosynthetic type (e.g., freshwater versus marine, C3 vs. C4 photosynthetic pathway). As with nitrogen isotopes, the heavier carbon isotope is also retained by consumers, though the effect is less pronounced with carbon isotopes. Together, carbon and nitrogen isotope ratios ($^{13}C/^{12}C$ and $^{15}N/^{14}N$ ratios, respectively) can provide insights into the trophic relationships and the reconstruction of species diets [31–33]. Similar, though less comprehensive, studies have investigated the trophic ecology of apple snails (Ampullariidae; [34]) and the reconstructed food webs from ecosystems they inhabit (*Pomacea* spp.; [35]). However, this study takes an integrative approach, using consumer and producer tissue samples, where (1) consumer diets are reconstructed using SIAR (Stable Isotope Analysis in R) and (2) PCR-mediated sequencing confirms all species identities, in native and non-native habitats. This approach is applicable to a plethora of invasive species, in a multitude of habitats, not only to *Pomacea canaliculata*.

This study sampled large numbers of *Pomacea canaliculata*, aquatic and riparian plant species, detritus, and seston or phyto and/or zoo plankton (where possible) to define the diet of *P. canaliculata* in native (Lakes Sauce and Dario, Maldonado, Uruguay) and non-native habitats (a Chinese lake and creek site in Hangzhou, Zhejiang, China, and Kawainui Marsh on Oahu, HI, USA).

## 2. Materials and Methods

### 2.1. Field Collection Sites

Native habitats included two sites within the native range of invasive apple snails (*Pomacea canaliculata*): (1) Lake Sauce, an unspoiled location, and (2) Lake Dario; the latter of which was altered by anthropogenic activities. Both sites were located in Maldonado, Uruguay; samples collected here were processed at the CURE (Central University of the Republic) in December of 2014. The first two non-native habitats were in China. These sites included (3) a Chinese lake site and (4) a creek site; both were located within XiXi National Park in the city of Hangzhou, Zhejiang, China. Samples collected here were later processed at the Zhejiang Provincial Key Laboratory of Biometrology and Inspection & Quarantine in July of 2017. The final non-native habitat was (5) Kawainui marsh, which is located on the island of Oahu, HI, USA. Samples collected here were processed at the Bernice Pauahi Bishop Museum, on Oahu, HI, in November of 2018. Exact coordinates for collection sites are listed in Appendix A.1, Study Collection Site Coordinates.

Lake Sauce was an unspoiled example of the natural freshwater habitat of *P. canaliculata*, teeming with thousands of apple snails, various macroinvertebrate species, aquatic and riparian plants, and more. Conversely, Lake Dario was a nearby anthropogenically disturbed site. The anthropogenic disturbance in Lake Dario was significant, as nearby businesses routinely removed freshwater macrophytes for recreation (e.g., fishing and/or boating). Additionally, in Lake Dario, the water levels were often modified via a culvert, and the native flora, around the periphery of the lake, were replaced with ornamental non-native plant species. The Chinese lake and creek sites and Kawainui Marsh, which were outside of the native range of *P. canaliculata*, with similar latitudes, had been invaded by *P. canaliculata*.

### 2.2. Pomacea canaliculata, *Aquatic and Riparian Plants, and Detritus Collections*

*Pomacea canaliculata* and plants were collected by hand or with nets. Collected materials were catalogued and stored frozen. Genetic samples from apple snails were stored in 95% ethanol, and surplus material, for stable isotope analyses, were rinsed with deionized water, weighed, and then dried (60 °C, 24 h). After drying, these samples were re-weighed, their dry mass was recorded, and they were macerated into a fine powder for isotope analysis. Plant samples were weighed and dried (60 °C, 24 h); re-weighed, the dry mass was recorded, and they were macerated into a fine powder for isotope analysis. Dried plant samples were utilized for isotope analysis and genetic barcoding.

Detritus (dead particulate organic material) was collected in 42-cm tall plastic cylinders (15 cm diameter). Cylinders were forced down into substrate and the contents pulled up and emptied into plastic bags before being filtered through a three-piece soil sieve set (1″, 0.2″, and 0.25″ mesh diameters). Detritus was dried at 60 °C for 24 h, weighed, and macerated into a fine powder for isotope analysis. Apple snail, plant, phyto and zooplankton and/or seston where available, detritus isotope analysis samples, and any corresponding genetic samples (in ethanol) were cold-stored (−20 °C) until being transported to Howard University for long-term storage (−80 °C). Tissue samples from apple snails used for genetic barcoding were taken from the mantle. The remainder of apple snail bodies (sans shells) were dried at 60 °C for 24 h, weighed, and macerated into a fine powder for isotope analysis.

### 2.3. Planktonic Organism Collection

Planktonic organisms were collected at Lakes (1) Sauce and (2) Dario in Maldonado, Uruguay, and (3) the Chinese lake site in Hangzhou, China. At Lakes (1) Sauce and (2) Dario (in Maldonado, Uruguay), planktonic organisms were collected using a mesh net tow and funneled through a Van Dorn bottle. A mesh filter permeable to 50 μm was used to collect zooplankton by size, and a filter permeable to 20 μm was used for phytoplankton. Collected material was rinsed in deionized water and transported to CURE. Samples were then condensed using vacuum filtration or by hand (with the aid of a dissection scope and tweezers) to separate phytoplankton, zooplankton, and seston from the liquid medium.

At (3) the Chinese lake site (in Hangzhou, Zhejiang, China) zooplankton and phytoplankton were collected simultaneously. Water from the mesh net tow was funneled through a section of PVC pipe (152.4 mm in length, 76.2 mm diameter) affixed with two successive internal mesh filters (50 μm and 20 μm). Collected material was rinsed with deionized water and transported to Zhejiang Provincial Key Laboratory of Biometrology and Inspection & Quarantine at Jiliang University. Samples were condensed using non-Silica syringe filters. All samples of planktonic organisms were oven-dried (60 °C, 24 h minimum) and stored at −20 °C before being transferred to Howard University for long-term storage at −80 °C.

### 2.4. Stable Isotope Analysis ($^{13}C$ and $^{15}N$)

Dried powders were weighed and loaded into tin cups and run on a Thermo Delta V Advantage mass spectrometer (Thermo Fisher Scientific (Bremen) GmbH, Bremen, Germany), coupled to a Costech 4010 Elemental Analyzer (EA) (Costech Analytical Technologies Inc., Valencia, CA, USA) or an Elementar Isotope Cube (USA) via a Thermo Conflo IV (USA) interface in continuous flow mode. All analyses were performed at the Smithsonian MCI Stable Isotope Mass Spectrometry Laboratory. Initial raw isotope values were calculated in Isodat 3.0 software. Raw values were subsequently calibrated using a 2-point linear correction. Reference materials for this correction included Urea-UIN-3 [36] and Costech acetanilide, both calibrated directly to USGS40 and USGS41 (L-glutamic acids). Weight %N and weight %C values were calculated using a peak area calibration from the homogenous Costech acetanilide. Isotope data are reported using standard delta notation: $\delta X = [(R_{sample} - R_{standard})/R_{standard}] \times 1000$, where X is the heavy isotope of interest ($^{15}N$ or $^{13}C$), R is the ratio of interest ($^{15}N/^{14}N$ or $^{13}C/^{12}C$), the standard is atmospheric air (N) or V-PDB (C), and units are per mil (‰). Stable isotope data are reported with an error of $\pm 0.2$‰, which is based on repeated measures of reference materials and samples. Weight %N and weight %C data are reported with an error of $\pm 0.5\%$.

### 2.5. SIAR (Stable Isotope Analysis in R)

To provide an even more resolute picture of the trophic ecology of *Pomacea canaliculata*, the Stable Isotope Analysis in R (SIAR; [37]), a mixing model [38], was selected to examine the contribution of available food resources to the diet of *P. canaliculata*, e.g., [39]. SIAR was selected, as it takes into account the expected fractionation of both $\delta^{15}N$ (2.98‰ $\pm$ 0.11 SD) [40] and $\delta^{13}C$ (0.5‰ $\pm$ 0.13 SD) [41] to determine the composition, and relative proportion, that respective food resources contribute to consumer diets.

SIAR is a statistical package available in the R statistics program. SIAR, using Bayesian inference, overcomes hurdles other (mixing) models encounter analyzing stable isotope data (e.g., uncertainty, underdetermined systems, incorporating variation in input parameters, and external sources of variation, as outlined by Parnell et al. [37]). SIAR accomplishes this by incorporating more sources of variation and dietary resources, while simultaneously creating a variety of probability distributions for inferred diets. SIAR was employed to analyze $\delta^{13}C$ and $\delta^{15}N$ values for *Pomacea canaliculata*, riparian and aquatic plants, and detritus, as well as seston and zooplankton where applicable, to define the diet of *P. canaliculata* in the native (Maldonado, Uruguay) and non-native (Hangzhou, China, and Oahu, HI, USA) habitats of this species.

All detritus and invasive apple snail stable isotope data from respective collection sites are listed in Appendix A.4 tables. These tables specifically list all individual detritus and *Pomacea canaliculata* samples from each respective collection site alongside corresponding δ15N and δ13C values used to construct SIAR models.

Appendix A.4 also lists the mean and standard deviation for corresponding δ15N and δ13C values recorded for all food resources in subsequent tables. These resources specifically were: aquatic and terrestrial (riparian) plants, plankton, detritus, phytoplankton, zooplankton, and seston used in SIAR models for all collection sites.

### 2.6. Plant Genetic Barcoding

Plant DNA Extraction for Uruguayan and Chinese Plant Species

Plant genomic DNA extractions were performed using an MP Fast Prep—24 Kit. PA2 Buffer was heated to 65 °C, and RNAse was set on ice. Plant genetic samples were placed in 2-mL screw-cap tubes, with ceramic spheres from the MP Fast Prep—24 Kit. The 2-mL screw-cap tubes were placed into the tissue homogenizer, and the MP Fast Prep—24 machine; which was set to a speed of 6.0 m/s, with the adapter set to "Quick prep".

Afterwards, Lysing Matrix A was selected on the MP Fast Prep—24 machine, the cycle was set to one, and the pause time was set to 300 s. Plants samples were homogenized for 30 s repeatedly, until pulverized. Afterwards, tubes were centrifuged at $10,000\times g$, for thirty seconds. Plant samples were removed from the centrifuge, and 300 µL of Buffer PA2 (cell lysis buffer) was added to each. Next, 10 µL of RNAse A was added to each sample, and they were inverted until mixed and centrifuged again ($10,000\times g$ for thirty seconds). Plant samples were incubated for ten minutes at 65 °C. Finally, Bioline Isolate II protocols (facilitating the precipitation, binding, washing, and elution of plant genomic DNA) were followed.

### 2.7. Hawaiian Plant Species Identifications

Plant species collected in Hawaii were identified using illustrations found in the Hawaii Wetland Field Guide [42] (pp. 249–263).

### 2.8. Pomacea canaliculata Genetic Barcoding

Total genomic DNA was extracted from all *Pomacea canaliculata* genetic samples using the QIAGEN DNeasy Blood and Tissue Kit (Cat No./ID: 69581). The QIAGEN Supplementary Protocol "Purification of total DNA from insects using the DNeasy Blood & Tissue Kit" was used for all *P. canaliculata* specimens. *P. canaliculata* genetic samples, with a maximum mass of 50 mg of tissue (absent dark pigmentary proteins, which inhibit the DNA extraction process), were placed into sterile labeled 1.5 mL centrifuge tubes (and subsequently macerated using individual micropestles to prevent cross-contamination), along with 180 µL of Buffer ATL and 20 µL of Proteinase K. The contents were vortexed thoroughly and incubated at 56 °C for 1 to 3 h or until all tissue therein had been completely dissolved. Afterwards, 200 µL of Buffer AL was added to the contents and vortexed again, and then 200 µL of cold Ethanol (0 °C) was added. The contents, now greater than 600 µL in total, were transferred via pipette to DNeasy Mini Spin column, set within a 2 mL collection tube.

These columns were centrifuged at $\geq6000\times g$ (8000 rpm) for one minute, and the flow through was discarded. Next, 500 µL of Buffer AW1 was pipetted into each column and centrifuged, again at $\geq6000\times g$ (8000 rpm) for one minute, and the flow through was again discarded. Afterwards, 500 µL of AW2 was pipetted into each column and centrifuged, at $\geq20,000\times g$ (14,000 rpm) for three minutes, and the flow through was discarded a third time. DNA extraction columns were transferred to labeled 1.5 mL centrifuge tubes, and 50 µL of Elution Buffer was pipetted into each and centrifuged at $\geq6000\times g$ (8000 rpm) for one minute. The prior step (involving elution buffer) was performed a second time, in a second labeled 1.5 mL centrifuge tube. Both genomic DNA samples (50 µL each) were stored in the Howard University Interdisciplinary Research Building (HUIRB) at $-80$ °C.

All aquatic and terrestrial plants, plankton, detritus, phytoplankton, zooplankton, and/or seston species identities (where applicable), along with the means by which said identifications were made, within respective collection sites are listed in Appendix A.5.

### 2.9. Polymerase Chain Reaction (PCR) Amplification

Genomic DNA samples from *Pomacea canaliculata* were utilized in polymerase chain reactions (as template DNA) with Bioline Company (USA) molecular reagents. Each PCR reaction had a total volume of 25 µL, composed of diH20 (9.25 µL), Bioline 5X Buffer (No Mg, 5 µL), $MgCl_2$ (2.5 mM, 1.25 µL), dNTPs (0.2 mM, 4 µL), a forward primer (0.15 µM, 1 µL), a reverse primer (0.15 µM, 1 µL), BSA (0.4 µg/µL, 1 µL), DMSO ([0.5%], 0.125 µL), Mango Taq (5 U/µL, 0.2 µL), and template DNA (2 µL).

Extracted genomic plant DNAs were amplified via PCR targeting the rbcl gene region, using the primers (1) rbcLF and (2) rbcLR. A master mix was prepared containing a total volume of 15 μL for each amplification, specifically composed of (1) 7.5 μL of 2× MyTaq Red Mix, (2) 2 μL of each primer (rbcLF and rbcLR), (3) 2.5 μL of deionized water, and (4) 1 μL of template DNA from each individual sample. A previously amplified sample of plant DNA (T-93) was used as a positive control, and negative control was deionized water in lieu of template DNA for each set of PCRs run. PCR tubes were then centrifuged again, ensuring each 15 μL volume contacted the bottom of corresponding PCR tubes, and placed in the thermocycler, and finally, the PCR program for the designated gene being amplified was run. PCR products were stored at 4 °C and subsequently sent out for sequencing.

### 2.10. Thermocycler

A Gene Touch PCR thermocycler with two-bay 96-well heat block was used to run all PCR programs. The mitochondrial cytochrome *c* oxidase subunit was targeted for *Pomacea canaliculata* to barcode all fauna [43] and compare them against available barcode databases using Geneious Prime 2019 (e.g., NCBI). The Cytochrome Oxidase Subunit I (COI) and Ribulose Bisphosphate Carboxylase Large (rbcL) gene primers, specific thermocycler programs, and the paired primers used for DNA amplifications are also described in Appendices A.2 and A.3; as are the respective primers, specific thermocycler programs, and paired primers utilized in this study.

### 2.11. Pomacea canaliculata Agarose Gels

Two percent Agarose gels were prepared by weighing 0.7 g of AMRESCO AGAROSE RA (Catalogue #:N605-500G), placing it into an Erlenmeyer flask with 35 mL of 1× SBE Buffer and 5 μL of Ethidium Bromide (EtBr) solution (10 μM), and heating in a microwave for one minute until boiling.

Once the 2% Agarose gel solidified, the gel was submerged beneath 1× BBE buffer, and 5 μL of Bioline Easy Ladder I, 3 μL of each PCR product, and 3 μL positive and negative controls were pipetted into the wells and electrophoresed at 140 V for ten minutes. Agarose gels were fluoresced using a UV trans-illuminator (USA) for COI (Cytochrome Oxidase Subunit I) PCR products and photographed. Successfully amplified PCR products were shipped to Eurofins Genomics for sequencing in a Eurofins Genomics 96-Well Prepaid (Eurofins Genomiv, Louisville, KY, USA) kit for crude PCR products. The resultant sequences were edited and identified using the Geneious Prime (2019 version) (New Zealand) and NCBI (National Center for Biotechnology Information (Rockville, MA, USA)) database tools therein.

### 2.12. Plant Agarose Gels

One percent Agarose gels were prepared with 0.5 g of AMRESCO AGAROSE RA (Catalogue #:N605-500G), 50 mL of 1× TBE Buffer, and 1 μL of Ethidium Bromide (EtBr) solution (10 μM). Once the 1% Agarose gel solidified, it was submerged in 1× TBE buffer, and 5 μL of Bioline Easy Ladder I, 3 μL of plant PCR products, and 3 μL of positive and negative controls were pipetted into the Agarose gel. Gels were electrophoresed at 100 V (for 25 min). Gels were fluoresced using a UV trans-illuminator; amplified DNA fragments (rbcL gene) were visualized and photographed. Successfully amplified PCR products were shipped to Eurofins Genomics for sequencing in a Eurofins Genomics 96-Well Prepaid kit for crude PCR products. The resultant sequences were identified via BLAST.

## 3. Results

### 3.1. Lake Sauce (Maldonado, Uruguay)

A total of thirteen plant species (seven aquatic plant species, six terrestrial plant species) were collected from Lake Sauce in Maldonado, Uruguay. The confirmed species identities of these plant species are listed in Table 1. Eleven plant species in total were identified to genus or species via sequencing data from a plastid locus (*rbcL*) and comparisons to available barcode databases (e.g., NCBI).

The stable isotope data for *Pomacea canaliculata*, plant species, zooplankton, and detritus collected in Lake Sauce defined between 59.2% and 77.7% of the diet of *P. canaliculata* therein (Figure 1; full list of stable isotope ($\delta^{15}$N and $\delta^{13}$C) values for individual *P. canaliculata* and detritus samples taken from collection sites is provided in the Appendix A data; additionally, a second list of the means and standard errors for the stable isotope values ($\delta^{15}$N and $\delta^{13}$C) for all food resources used in SIAR models is provided in Table 2 for each respective collection site). This range for the percentage of the invasive apple snail diet represents a 95% confidence interval. The interval in this case does not overlap with 100%. This is because, despite extensive sampling, the food resources utilized in the SIAR model did not represent the entirety of the apple snail diets therein. This diet was composed primarily of four terrestrial plant species (*Persicaria* sp., *Breynia* sp., *Hydrocotyle* sp., and *Rumex* sp.) and one aquatic species (*Salvinia* sp.); together, they represented some 49.7% to 62.4% of the diet of *P. canaliculata*. Zooplankton represented between 2.9% and 3.8%, and all other food resources together represented between 6.0% and 10.7% of the diet of *P. canaliculata* in Lake Sauce.

**Table 1.** List of all food resources available in Lake Sauce, in Maldonado, Uruguay; AP—aquatic plant; TP—terrestrial plant.

| Plants/Plankton/ Detritus | Sample Size (n) | Species ID | Source of Identification |
|---|---|---|---|
| AP1 | 5 | *Certophyllum* sp. | Sequence Data |
| AP2 | 10 | *Salvinia* sp. | Sequence Data |
| AP3 | 10 | *Egeria* sp. | Sequence Data |
| AP4 | 10 | Monocot Perennial | Morphological Characteristics |
| AP5 | 10 | *Ludwigia* sp. | Sequence Data |
| AP6 | 10 | *Eichhornia crassipes* | Sequence Data |
| AP7 | 10 | *Echinodorus* sp. | Sequence Data |
| TP1 | 10 | *Poaceae* sp. | Sequence Data |
| TP2 | 5 | *Persicaria* sp. | Sequence Data |
| TP3 | 10 | *Breynia* sp. | Sequence Data |
| TP4 | 10 | *Hydrocotyle* sp. | Sequence Data |
| TP5 | 10 | *Rumex* sp. | Sequence Data |
| TP6 | 5 | **Not Identified** | N/A |
| Detritus | 5 | N/A | N/A |
| Zooplankton | 6 | N/A | N/A |

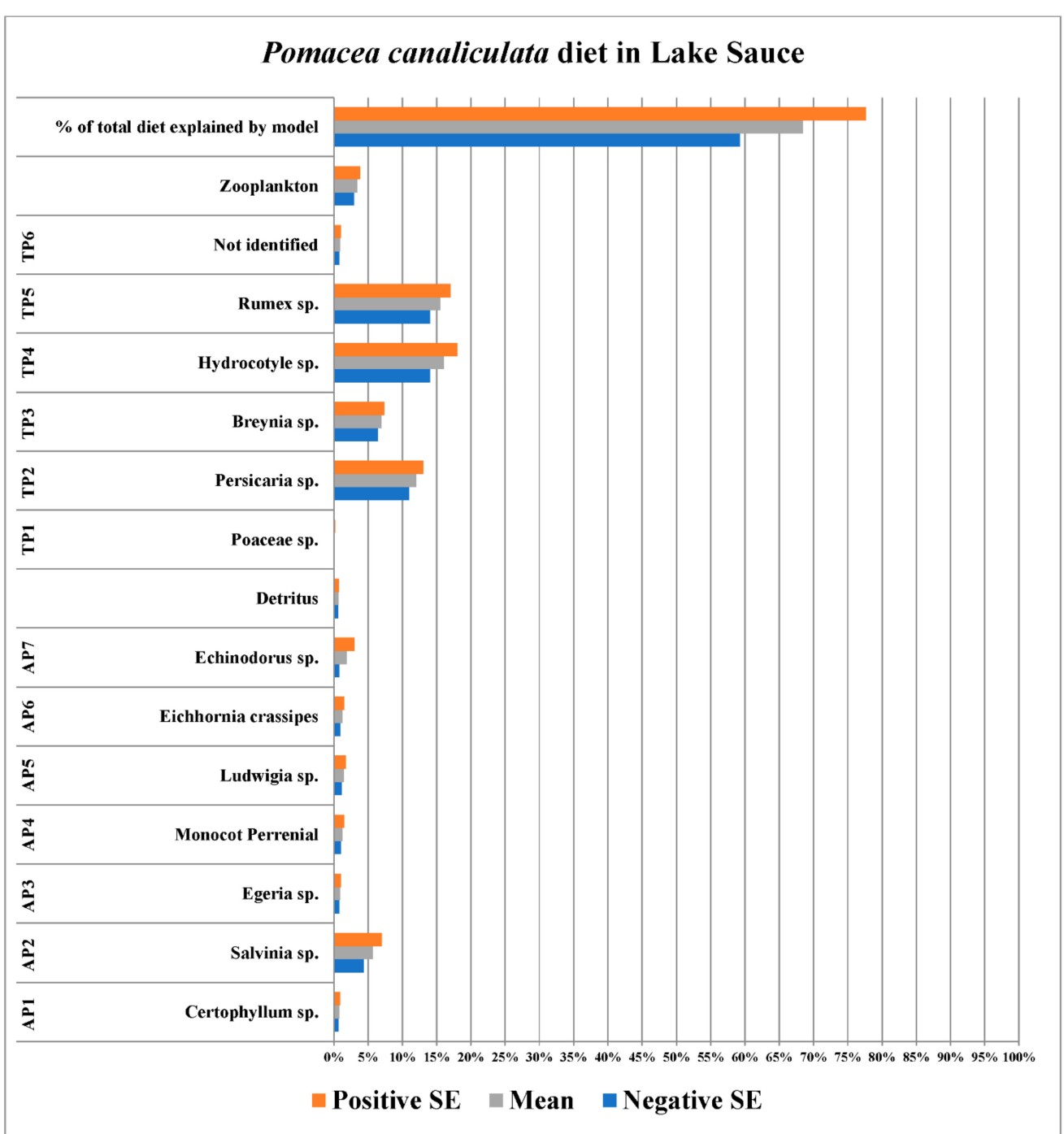

**Figure 1.** Diet of *Pomacea canaliculata* in Lake Sauce, Maldonado, Uruguay. AP and TP acronym refers to aquatic and terrestrial plant species, respectively.

### 3.2. Lake Dario (Maldonado, Uruguay)

A total of nine plant species (seven aquatic plant species and two terrestrial pant species) were collected from Lake Dario in Maldonado, Uruguay. The confirmed species identities of these plant species are listed in Table 2. All nine plant species were identified to genus and/or species via sequencing data from a plastid locus (*rbcL*) and comparisons made with available barcode databases (e.g., NCBI).

The stable isotope data for *Pomacea canaliculata*, plant species, zooplankton, seston, and detritus collected in Lake Sauce defined between 78.4% and 89.5% of the diet of *P. canaliculata* therein (Figure 2; full list of stable isotope ($\delta^{15}$N and $\delta^{13}$C) values for individual *P. canaliculata* and detritus samples taken from collection sites is provided in the Appendix A data; additionally, a second list of the means and standard errors for the stable isotope values ($\delta^{15}$N and $\delta^{13}$C) for all food resources used in SIAR models is provided in Table 2 for each respective collection site). This diet was composed primarily of a filamentous alga (*Spirogyra* sp.), seston, and detritus; which represented between 74.0% and 83.0% of the diet of *P. canaliculata* therein. All other food resources represented between 4.4% and 6.5% of the diet of *P. canaliculata* in Lake Dario.

**Table 2.** List of all food resources available in Lake Dario, in Maldonado, Uruguay; AP—aquatic plant; TP—terrestrial plant.

| Plants/Plankton/ Seston/Detritus | Sample Size (n) | Species Identification | Source of Identification |
|---|---|---|---|
| AP1 | 10 | *Salvinia* sp. | Sequence Data |
| AP2 | 10 | *Ludwigia* sp. | Sequence Data |
| AP3 | 10 | *Egeria densa* | Sequence Data |
| AP4 | 10 | *Potamogeton* sp. | Sequence Data |
| AP5 | 10 | *Pistia stratiotes* | Sequence Data |
| AP6 | 10 | *Potamogeton* sp. | Sequence Data |
| AP7 | 10 | *Eichhornia Crassipes* | Sequence Data |
| AP8 | 2 | *Spirogyra* sp. (Filamentous Alga) | Microscopy |
| TP1 | 5 | *Poaceae* sp. | Sequence Data |
| TP2 | 10 | *Ludwigia* sp. | Sequence Data |
| Detritus | 5 | N/A | N/A |
| Zooplankton | 3 | N/A | N/A |
| Seston | 4 | N/A | N/A |

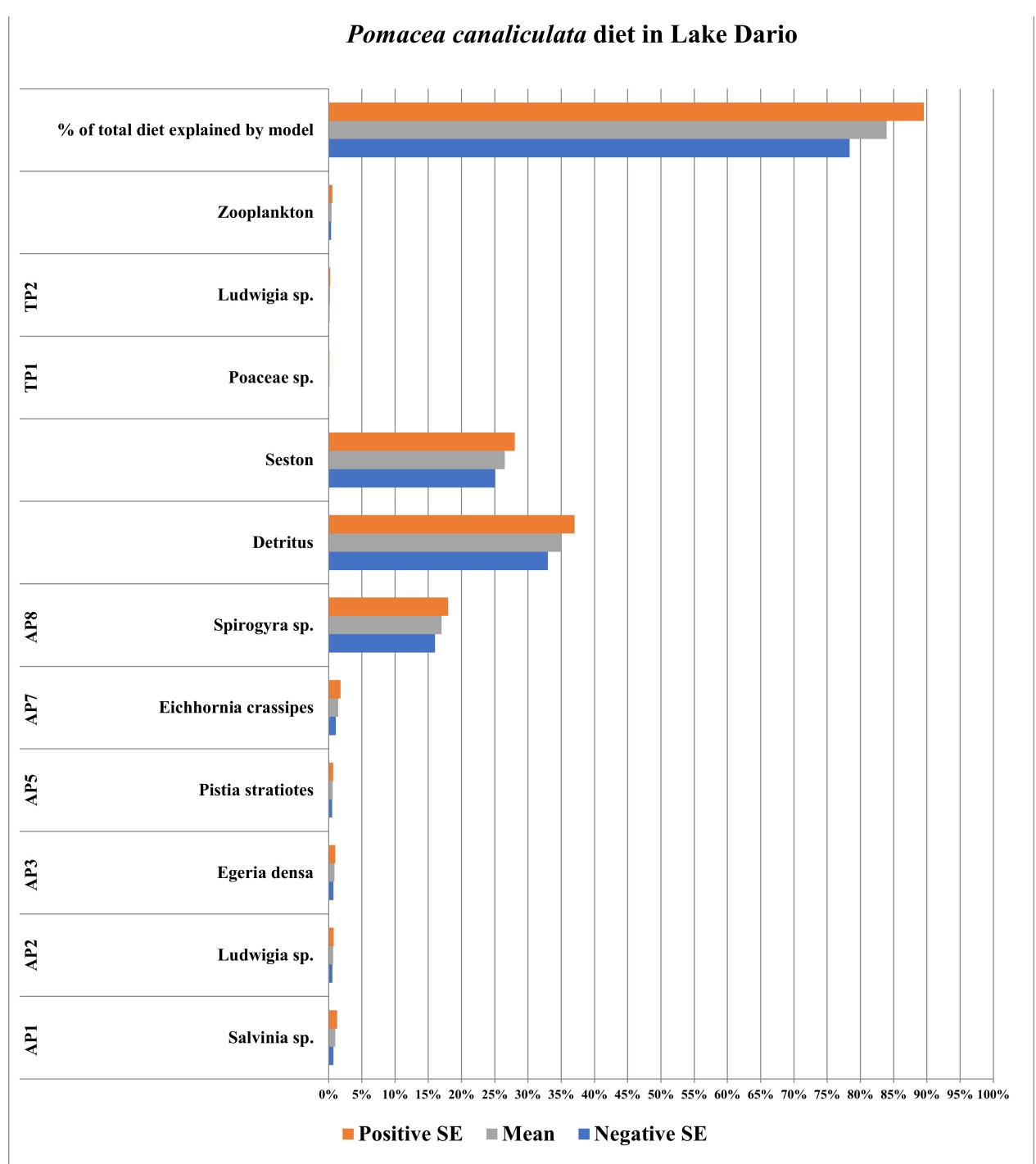

**Figure 2.** Diet of *Pomacea canaliculata* in Lake Dario, Maldonado, Uruguay. AP and TP acronym refers to aquatic and terrestrial plant species, respectively.

### 3.3. Chinese Lake Site (Hangzhou, Zhejiang, China)

A total of nine plant species (four aquatic plant species and five terrestrial plant species) were collected from a lake in XiXi National Park, Hangzhou, China. The confirmed species identities of plant species are listed in Table 3. Eight of nine plant species were identified to genus and/or species via sequencing data from a plastid locus (*rbcL*) and available barcode databases (e.g., NCBI).

The stable isotope data for *Pomacea canaliculata*, plant species, and detritus collected in the Chinese lake site defined between 83.3 and 92.2% of the diet of *P. canaliculata* therein (Figure 3; full list of stable isotope ($\delta^{15}$N and $\delta^{13}$C) values for individual *P. canaliculata* and detritus samples taken from collection sites is provided in the Appendix A data; additionally, a second list of the means and standard errors for the stable isotope values ($\delta^{15}$N and $\delta^{13}$C) for all food resources used in SIAR models are provided in Table 2 for each respective collection site). This diet was composed primarily of zooplankton and a single terrestrial plant (unidentified), which represented between 79.0% and 80.0% of the diet of *P. canaliculata* therein. Zooplankton alone constituted between 51.0% and 52.0%, and all other food resources represented between 4.3% and 8.2% of the diet of *P. canaliculata* in the Chinese lake site.

**Table 3.** List of all food resources available at the Chinese lake site, in Hangzhou, China; AP—aquatic plant; TP—terrestrial plant.

| Plants/ Detritus | Sample Size (n) | Species ID | Source of Identification |
|---|---|---|---|
| AP 1 | 10 | *Thalia* sp. | Sequence Data |
| AP2 | 10 | *Zizania* sp. | Sequence Data |
| AP3 | 10 | *Aternanthera* sp. | Sequence Data |
| AP4 | 5 | *Carex* sp. | Sequence Data |
| TP1 | 5 | *Glechoma* sp. | Sequence Data |
| TP2 | 5 | *Oplismenus* sp. | Sequence Data |
| TP3 | 5 | *Erigeron* sp. | Sequence Data |
| TP4 | 5 | **Not identified** | N/A |
| TP5 | 5 | *Nastus* sp. | Sequence Data |
| Detritus | 5 | N/A | N/A |

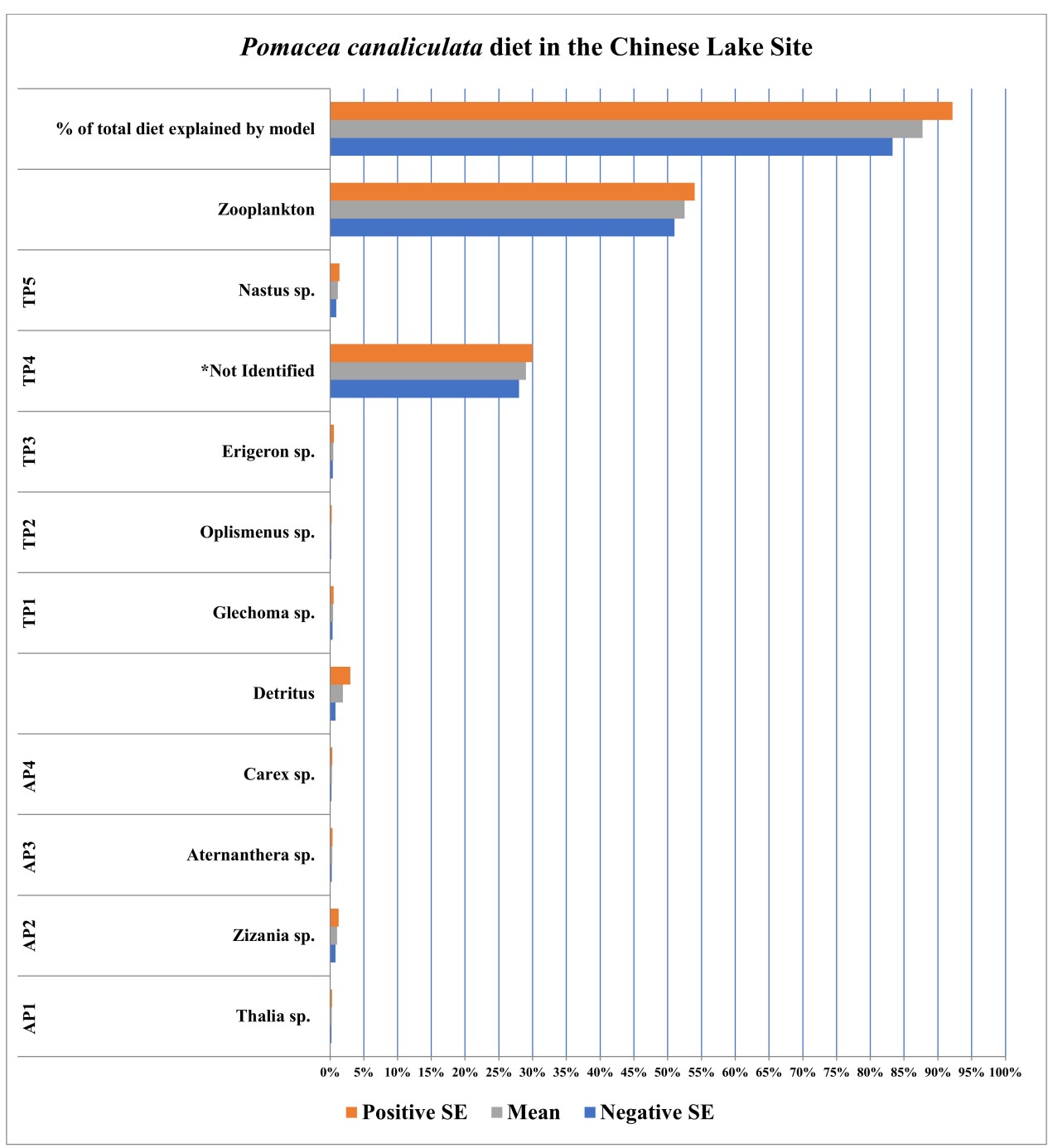

**Figure 3.** Diet of *Pomacea canaliculata* in the Chinese lake site; Hangzhou, China. AP and TP acronym refers to aquatic and terrestrial plant species, respectively.

*3.4. Chinese Creek Site (Hangzhou, China)*

Five plant species (three aquatic and two terrestrial plant species) were collected from the Chinese creek site in XiXi National Park Hangzhou, China. The confirmed species identities of plant species are listed in Table 4. All five plant species were identified to genus and/or species via sequencing data from one plastid locus (*rbcL*) and comparisons to available barcode databases (e.g., NCBI).

The stable isotope data for *Pomacea canaliculata*, plant species, and detritus collected in the Chinese creek site defined between 94.2 and 110.8% of the diet. This 95% confidence interval overlapped with 100%, meaning 100% of the diet of *P. canaliculata* was defined in this case (Figure 4; full list of stable isotope ($\delta^{15}$N and $\delta^{13}$C) values for individual *P. canaliculata* and detritus samples taken from collection sites is provided in the Appendix A data; additionally, a second list of the means and standard errors for the stable isotope values ($\delta^{15}$N and $\delta^{13}$C) for all food resources used in SIAR models is provided in Table 2 for each respective collection site). This diet was composed primarily of three aquatic plant species (*Hygrophila* sp., *Iris pseudacarus*, and *Oenathe* sp.), which represented between 72.0 and 76.0% of the diet of *P. canaliculata* therein. The two remaining terrestrial plant species represented between 21.0 and 27.0% of the diet of *P. canaliculata*, or 11.0 to 14.0% (*Pteris henryi*) and 10.0 to 13.0% (*Oplismenus* sp.), respectively, while detritus only represented between 1.2 and 1.8% of the diet of *P. canaliculata* in the Chinese creek site (Figure 4).

**Table 4.** List of all food resources available in the Chinese creek site, in Hangzhou, China; AP—aquatic plant; TP—terrestrial plant.

| Plants/Detritus | Sample Size (n) | Species | Source of Identification |
|:---:|:---:|:---:|:---:|
| AP1 | 10 | *Oenathe* sp. | Sequence Data |
| AP2 | 10 | *Hygrophila* sp. | Sequence Data |
| AP3 | 10 | *Iris pseudacarus* | Sequence Data |
| TP1 | 12 | *Pteris henryi* | Sequence Data |
| TP2 | 10 | *Oplismenus* sp. | Sequence Data |
| Detritus | 5 | N/A | N/A |

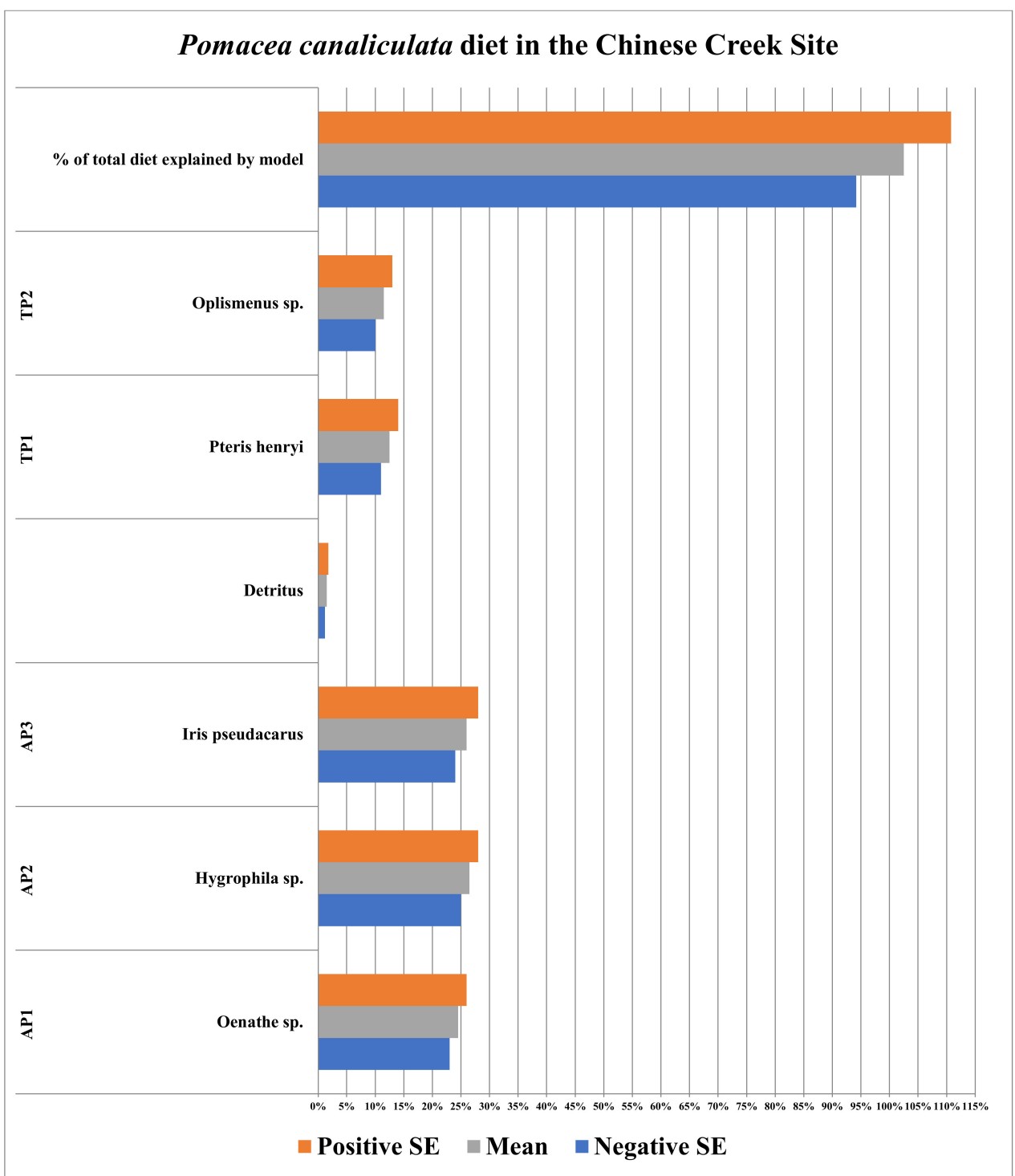

**Figure 4.** Diet of *Pomacea canaliculata* in the Chinese creek site in Hangzhou, China. AP and TP acronym refers to aquatic and terrestrial plant species, respectively.

### 3.5. Kawainui Marsh (Oahu, HI, USA)

Four plant species (three aquatic and one terrestrial plant species) were collected in Kawainui Marsh in Oahu, HI, USA. The confirmed species identities of plant species are listed in Table 5. All the four plant species were identified to genus and/or species using the Hawaii Wetland Field Guide, written by Erickson and Puttock (2006), and information on pages 249–263.

The stable isotope data for *Pomacea canaliculata*, plant species, and detritus collected in the Kawainui Marsh defined between 93.2 and 100.8%. This 95% confidence interval overlapped with 100%, meaning 100% of the diet of *P. canaliculata* was defined in this case (Figure 5; full list of stable isotope ($\delta^{15}$N and $\delta^{13}$C) values for individual apple snails *P. canaliculata* and detritus samples taken from collection sites is provided in the Appendix A data; additionally, a second list of the means and standard errors for the stable isotope values ($\delta^{15}$N and $\delta^{13}$C) for all food resources used in SIAR models is provided in Table 2 for each respective collection site). This diet was composed primarily of detritus (58.0 to 61.0%) and a single aquatic plant species (*Bacopa monnieri*; 33–35%), which together represented between 91.0 and 96.0% of the diet of *P. canaliculata* therein. All other food resources, a terrestrial and two aquatic plant species together (*Pteris henryi, Iris pseudacarus*, and *Hygrophila* sp. respectively), represented only 2.2 to 4.9% of the diet of *P. canaliculata* in the Kawainui Marsh (Figure 5).

**Table 5.** List of all food resources available in Kawainui Marsh, Oahu, HI, USA; AP—aquatic plant; TP—terrestrial plant.

| Plants/Detritus | Sample Size (n) | Species | Source of Identification |
|---|---|---|---|
| AP1 | 5 | *Bacopa monnieri* | Field Identification |
| AP2 | 5 | *Brahchiaria mutica* | Field Identification |
| AP3 | 5 | *Echinochloa crusgalli* | Field Identification |
| TP1 | 5 | *Cynodon dactylon* | Field Identification |
| Detritus | 5 | N/A | N/A |

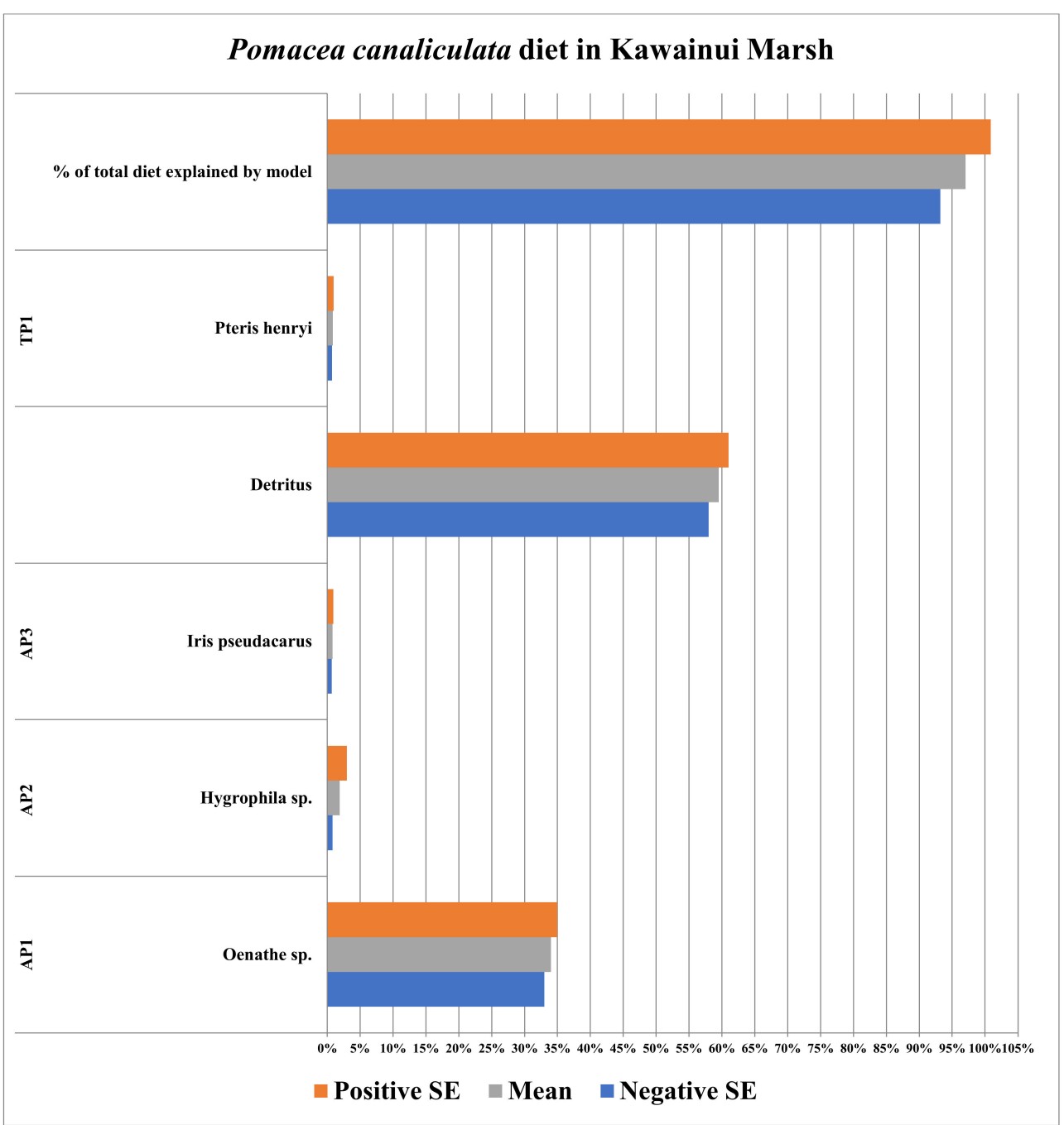

**Figure 5.** Diet of *Pomacea canaliculata* in Kawainui Marsh, Oahu, HI, USA. AP and TP acronym refers to aquatic and terrestrial plant species, respectively.

## 4. Discussion

The stable isotope data collected from all native and non-native collection sites, and the models subsequently produced from them, have defined the diet of *Pomacea canaliculata* in native versus non-native habitats. The previously stated hypothesis is that the diet of *P. canaliculata* experiences shifts in trophic ecology and/or feeding behavior when moving from undisturbed native to (1) disturbed native and/or (2) non-native habitats. Specifically, this hypothesis predicts that this trophic shift will involve a shift from a generalist (broad) to a more constrained (or specialized) diet, a prediction supported by the study results in many ways enumerated here.

In undisturbed, unspoiled, portions of the native range of *P. canaliculata* the species displays a generalist feeding strategy, as hypothesized, with a diet that incorporates a variety of food resources, with each dietary component representing a smaller proportion of the overall diet when juxtaposed with diets in subsequently sampled habitats. Stated more explicitly, a greater number of available food resources are utilized in the apple snail diet modeled for the pristine portions of the native habitat (Lake Sauce). In the SIAR model produced from stable isotope data collected in Lake Sauce, no utilized food resource represents greater than 18% of the overall diet (Figure 1). These results provided increasingly stark contrast when juxtaposed with the ever more specialized diets of *P. canaliculata*, as depicted by SIAR models, in (1) the anthropogenically disturbed portion of the native range (Lake Dario, Figure 2) and, subsequently, (2) the three non-native habitats (the Chinese lake and creek site, Figures 3 and 4; and Kawainui Marsh, Figure 5) studied.

The anthropogenic disturbance in the second study site within the native range of *P. canaliculata* (Lake Dario) was significant. This translated to a loss of productivity via the periodic and habitual removal of freshwater macrophytes to make space for human recreational activities, causing the unavailability of native aquatic plant species (as food resources) in this site. This might explain the shift of *Pomacea canaliculata* diets within the native range (observed between Lake Sauce and Lake Dario), where more readily available food resources (in Lake Dario) are unaffected (if not made more readily available) by the removal of macrophytes (e.g., detritus, seston, and a filamentous alga). These more readily available food resources, combined, represented between 74% and 83% of the overall apple snail diet in Lake Dario (Figure 2).

This shift represents a change from the generalist diet and/or feeding behavior of *Pomacea canaliculata* in Lake Sauce (Figure 1) to a more constrained diet, feeding on a subset of food resources based on their availability. This constrained (more specialized) diet falls in line not only with the observed anthropogenic impacts on Lake Dario but also the increased percentage of *P. canaliculata* diets defined by SIAR models from Lake Sauce versus Lake Dario (68.5 ± 9.2% versus 89.5 ± 5.6%, in Lake Sauce versus Lake Dario, respectively, Figures 1 and 2). Furthermore, the proportion of the three most utilized food resources is greater in Lake Dario (Figure 2) than that of the single most utilized food resource in *P. canaliculata* diets discerned in Lake Sauce (at 18%, Figure 1). These results from the native habitats support the aforementioned hypothesis of trophic shifts this study set out to test.

In the Chinese lake site, this trend of diet constriction (specialization) continued as the diet of *Pomacea canaliculata* therein became even more specialized that in both native study sites (Figure 3 versus Figures 1 and 2 respectively). Here (in the Chinese lake site), the percentage of *P. canaliculata* diets defined by SIAR models in the Chinese lake site versus the anthropogenically disturbed Lake Dario also increases slightly; though the difference between the means is statistically indistinct (from 89.5 ± 5.6% to 92.2 ± 4.5% in Lake Dario versus the Chinese lake site, respectively, illustrated in Figures 2 and 3). This difference, though not statistically significant, does indicate that the diet of invasive apple snails is indeed becoming increasingly constricted (more specialized) in the move from the native to non-native habitats, as the portion of the overall diet explained by SIAR models increases.

Additionally, the most utilized food resource in the diet of *Pomacea canaliculata* in the Chinese lake site (zooplankton) represents greater than 50% of the overall diet (Figure 3). For juxtaposition, the three most utilized food resources in *P. canaliculata* diets in Lake Dario (filamentous algae, seston, and detritus) represented only 18 ± 1%, 28 ± 1.5%, and 37 ± 2%, respectively. These data indicate the diet and/or feeding behavior of *P. canaliculata* shift from a generalist diet, with a broad array of food resources being utilized while individually representing smaller proportions of the overall diet, to a more constricted (specialized) diet, where the number of food resources utilized and the proportion each represents in the overall diet are greater. This is evident when juxtaposing apple snail diets (1) between undisturbed (Lake Sauce, Figure 1) and anthropogenically disturbed portions of the native range (Lake Dario, Figure 2), but also (2) between native and non-native habitats (from Lake Sauce to Lake Dario and the Chinese lake site, as illustrated by Figures 1–3).

Data from the Chinese creek site did not immediately fit the pattern of fewer food resources being utilized by invasive apple snails, as a greater number of food resources were utilized by *Pomacea canaliculata,* which represented smaller proportions of the overall diet than in the Chinese lake site (Figures 3 and 4). However, the percentage of *P. canaliculata* diets defined by SIAR models in the Chinese lake site versus the Chinese creek site did increase (from $92.2 \pm 4.5\%$ to $102.5 \pm 8.3\%$ in the Chinese lake versus creek sites, respectively, Figures 3 and 4); though this difference was, again, not statistically significant.

One anecdotal observation made in the Chinese creek site was the decreased environmental heterogeneity, relative to the Chinese lake site. A second observation, based on the plant species catalogued therein, was decreased plant diversity between the Chinese lake and creek sites (Tables 3 and 4). The diet of *Pomacea canaliculata* in the Chinese creek site was more general (Figure 4) than in the Chinese lake site (Figure 3), but when compared to unspoiled portions of the native range of *P. canaliculata* (Lake Sauce, Figure 1), stark differences as to which food resources are utilized between these sites (Lake Sauce and the Chinese creek site) become obvious. In Lake Sauce (Figure 1), the food resources utilized were predominantly terrestrial plants, while in the Chinese creek site, freshwater macrophytes were utilized (Figure 4). This shift in diet and feeding behavior (between Lake Sauce and the Chinese creek site), from terrestrial plants to freshwater macrophytes as utilized food resources, could provide a mechanism facilitating the adverse ecological impacts of *P. canaliculata* (the removal of freshwater macrophytes and the shifting of freshwater habitats to microbially dominated, eutrophic, conditions) in non-native habitats. In the Chinese lake site (Figure 3), a terrestrial plant species represented the second largest component of apple snail diets therein, representing some 40% of the overall diet. Perhaps differences in the nutritional value of terrestrial food resources between Lake Sauce and the Chinese lake and creek sites would explain the differences in food resource utilization amongst them.

Data from Kawainui Marsh, Hawaii, also indicated the diet of *Pomacea canaliculata* was more constrained (specialized) than all other non-native collection sites with a single aquatic plant (*Oenathe* sp.) and detritus (together) comprising 91% to 96% of the overall diet (with the two food resources representing $34 \pm 1\%$ and $59.5 \pm 1.5\%$, respectively, of the overall diet of invasive apple snails therein, Figure 5). However, the percentage of *P. canaliculata* diets defined by SIAR models in the Chinese creek site versus Kawainui Marsh decreased slightly, though again, these means are statistically indistinct (with the means being $102.5 \pm 8.3\%$ and $97 \pm 3.8\%$, respectively). Nonetheless, both intervals overlapped with 100%.

In Kawainui Marsh, *P. canaliculata* diets were constricted or specialized (Figure 5) when juxtaposed with corresponding diets in unspoiled portions of the natural range of *P. canaliculata* (Lake Sauce; Figure 1). *P. canaliculata* diets in Kawainui Marsh utilized aquatic food resources, as observed in the Chinese creek site (Figure 4), and those utilized food resources (aquatic plants) represented 34% of *P. canaliculata* diets therein. However, detritus represented some 60% of overall diet in Kawainui Marsh (Figure 5).

SIAR models demonstrate that *Pomacea canaliculata* diets in native and non-native habitats vary broadly. They also illustrate that *P. canaliculata* experience food resource limitations in anthropogenically disturbed and non-native habitats. A greater number of all available food resources, particularly terrestrial plant resources, are readily utilized by *P. canaliculata* in unspoiled and highly productive portions of the native range (Lake Sauce). Furthermore, significantly, these utilized food resources represent a more evenly distributed proportion of the overall diet of *P. canaliculata*. Increasingly, in (1) anthropogenically disturbed sites (Lake Dario) within that same native range and (2) subsequently studied non-native habitats (the Chinese lake and creek sites and Kawainui Marsh), *P. canaliculata* diets become constricted and therefore more and more specialized.

Perhaps the shifting of *Pomacea canaliculata* diets is a result of a limited number of food resources, or alternatively a shortfall in the quality of these food resources. When *P. canaliculata* enter anthropogenically disturbed native habitats and/or non-native habitats, coevolved predatory species such as limpkins and snail kites [44,45], as well as coevolved

competitor populations, are absent. This allows apple snails to grow under reduced selection pressure, and their feeding behaviors and/or diet changes alongside this modified intraspecific and/or interspecific competition [46]. Additionally, in natural levels of production, movement between undisturbed parts of the native range of *P. canaliculata* (within the Pantanal), one of the most productive wetland habitats on Earth, and anthropogenically disturbed habitats therein and/or non-native habitats may force *P. canaliculata* to modify its diet to fulfill nutritional requirements as predicted by the optimal foraging theory [47]. This idea is supported by the stark differences in plant diversity amongst native (Tables 1 and 2) versus non-native habitats (Tables 3–5), bolstering the idea that reduced production in anthropogenically disturbed native and/or non-native habitats reduces the number and availability of food resources for *P. canaliculata*. In any case, the SIAR models constructed from data acquired in this study support the hypothesis that shifts in the trophic ecology and/or feeding behavior of invasive apple snails (*P. canaliculata*) exist, and furthermore, that diets become increasingly specialized in the move from undisturbed native habitats to (1) anthropogenically disturbed portions of the native habitat and (2) non-native habitats.

The proportions of different plant species included in the diet of *P. canaliculata* in anthropogenically disturbed native and/or non-native habitats may be predicted in the future; provided additional collection sites are investigated utilizing stable isotopes and corresponding SIAR models (see Parnell et al. [37] and Phillips et al. [38]). The methods employed in this study provide a blueprint by which long-term stable isotope studies may continue and provide insights as to: (1) the susceptibility of non-native habitats to biotic invasion based on the availability of utilizable food resources, (2) the adverse ecological impacts on ecosystem function and/or food web stability that may follow if invasion occurs, (3) the identity of producer species that may be unpalatable due to the presence of secondary metabolites and have possible use as a biotic control of *P. canaliculata* or other invasive species, and (4) the preservation, conservation, and/or restoration of terrestrial, marine, and aquatic habitats, provided stable isotope data were collected before biological invasions or anthropogenic disturbances occur.

## 5. Conclusions

It may seem obvious that the impacts of invasive species, including *Pomacea canaliculata*, on communities vary both temporally and spatially [48]. Nonetheless, mechanisms that explain the extirpation of non-native species from some portions of habitats, but not others, are poorly understood. The community assembly theory states that functionally similar species (e.g., species that occupy the same ecological niche) should not be able to coexist (assembly theory; [49]), and instances where native species have experienced precipitous declines in population due to direct competition with functionally similar invasive species abound [50,51]. Similarly, introduced and/or invasive species have been extirpated by more competitive, newly arrived, invaders [52,53]. However, evidence suggests that competitive exclusion of native species by biological invaders is rare in freshwater habitats [54–57].

However, as dietary shifts for *Pomacea canaliculata* are shown here between native and non-native habitats, it is shown that competitive exclusion of native species (be they herbivore, omnivores, or indirectly carnivores) is real for invasive species. The study of invaders' trophic ecology, within (a) native and (b) non-native habitats, and (c) knowledge of the natural trophic ecology of these non-native habitats (absent invaders) can illustrate the ecological mechanisms (1) facilitating successful invasions, (2) determining the distribution of invaders and native species, and (3) determining the intensity and types of competition they experience temporally and spatially.

**Author Contributions:** K.E.S.II conceived project design and design methodologies, completed all fieldwork, prepared samples, analyzed samples, processed data, analyzed data, and wrote and edited all manuscript drafts. C.A.M.F. assisted with project design and methodologies, provided training, provided laboratory support, analyzed samples, assisted with data analysis, and edited all manuscript drafts. F.L.C.J. assisted with project design and methodologies, assisted with general project management, provided funding and laboratory space, assisted with data analysis, and edited all manuscript drafts. All authors have read and agreed to the published version of the manuscript.

**Funding:** Financial support in the form of a $1870.00 grant, specifically the 2018 Frederic Weiss Memorial Award, was provided by the Conchologist of America in support of the portion of this research conducted on Oahu, HI, USA. In addition, the APC (Article Publication Charges) for this manuscript were covered entirely by the Department of Environmental Science at The University of Arizona in Tucson, AZ, USA.

**Institutional Review Board Statement:** Ethical review and approval were waived for this study due to the fact this study solely involved the handling of invertebrate and plant species, as well as planktonic organisms, by K.E.S.II. As such, the IACUC (Institutional Animal Care and Use Committee) at Howard University and the Howard University Graduate School exempted this project from IACUC review.

**Informed Consent Statement:** Not applicable as this study did involve humans.

**Data Availability Statement:** Data supporting reported results can be found in the Appendix A. Sequence data used for invasive apple snail species identifications are held by K.E.S.II.

**Acknowledgments:** I acknowledge Leslie Ries, in the Department of Biology at Georgetown University, for her support in completing my field work and the opportunity to share my research endeavors and receive critical and valuable feedback from her laboratory; undergraduate students from Howard University (Zahra Mansur and Brittany Galloway) for their help in completing my stable isotope analyses; the CURE Institute in Maldonado, Uruguay, Jiliang University in Hangzhou, China, Howard University's Department of Biology, The Bishop Museum on Oahu, HI, and The Smithsonian Institution for support to complete fieldwork for this research. Last, I acknowledge funding from the Conchologist of America in support of this research in Hawaii.

**Conflicts of Interest:** There are no conflicts of interest, financial or otherwise, that have influenced authors' objectivity towards the production and/or publication of this research.

## Appendix A

*Appendix A.1. Study Collection Site Coordinates*

**Uruguayan Collection Sites:**
 **Site #1** (Lake Sauce) Laguna Del Sauce (34°50′59.78″ S, 55°3′0.98″ W)
 **Site #2** (Lake Dario) Laguna Del Diario (34°53′59.7″ S, 55°00′29.3″ W) near Maldonado, Uruguay.
**Chinese Collection Sites:**
 **Site #3** (Chinese Creek Site, 30°16′21.3″ N, 120°03′45.4″ E) in XiXi National Park, Hangzhou, Zhejiang, China.
 **Site #4** (Chinese Lake Site, 30°16′26.5″ N, 120°03′47.4″ E) in XiXi National Park, Hangzhou, Zhejiang, China.
**Hawaiian Collection Site:**
 **Site #5** (Kawainui Marsh Site, 21°23′02.9″ N, 157°45′37.3″ W)
 Location was 200 feet east of the Maunawilli Stream within Kawainui Marsh, Oahu, HI, USA.

*Appendix A.2. Respective Primers Used in This Study*

**The Cytochrome Oxidase Subunit I (COI) primers used were:**

    **LCO1490 Universal Arthropod Primer** (Brandon-Mong et al., 2015; [58])

    (5′-GTTAAAAAAAACCGGAACCGGGGCCCCAAGGTTGGTCAACAAATCATAAAGATATTGG-3′)

    **COXAR Caenogastropoda Primer** (Colgan et al., 2001; [59])

    (5′-ATATAWACTTCWGGGTGACC-3′)

    The rbcl primers used were:

1.   **rbcLaF** (Bafeel et al., 2011; [60])

    (5′-ATGTCACCACAAACAGAGACTAAAGC-3′)

2.   **rbcLaR**, (Bafeel et al., 2011; [60])

    (5′-GTAAAATCAAGTCCACCRCG-3′)

*Appendix A.3. Specific Thermocycler Programs and Paired Primers Utilized in This Study*

    Paired primers and associated thermocycle for DNA amplifications:

(1)   *Pomacea canaliculata*

    **Primers: LCO1490 and COXAR** (Colgan et al., 2001; [59])

| 5 min | 1 min | 1 min | 30 s | 30 s | 45 s | 10 min |
|-------|-------|-------|------|------|------|--------|
| 95 °C | 45 °C | 72 °C | 95 °C | 48 °C | 72 °C | 4 °C |
| 75 Cycles | | | | | | |

(2)   **Plant Species**

    **Primers: rbcLaF and rbcLaR** (Bafeel et al., 2011; [60])

| 5 min | 1 min | 1 min | 30 s | 30 s | 45 s | 10 min |
|-------|-------|-------|------|------|------|--------|
| 95 °C | 45 °C | 72 °C | 95 °C | 48 °C | 72 °C | 4 °C |
| 50 Cycles | | | | | | |

*Appendix A.4.*

Tables A1–A5: List of all individual detritus and apple snail (*Pomacea canaliculata*) samples from each respective collection site alongside corresponding $\delta^{15}N$ and $\delta^{13}C$ values used to construct SIAR models.

**Table A1.** Lake Sauce recorded individual stable isotope values ($\delta^{15}N$ and $\delta^{13}C$) for detritus and apple snails (*Pomacea canaliculata*).

| Sample | $\delta^{15}N$ | $\delta^{13}C$ |
|---|---|---|
| | (‰, Air) | (‰, VPDB) |
| *Pomacea canaliculata* | 5.0 | −27.0 |
| *Pomacea canaliculata* | 5.7 | −28.7 |
| *Pomacea canaliculata* | 4.3 | −25.7 |
| *Pomacea canaliculata* | 5.8 | −27.3 |
| *Pomacea canaliculata* | 5.6 | −28.1 |
| *Pomacea canaliculata* | 5.9 | −26.7 |
| *Pomacea canaliculata* | 5.0 | −26.7 |
| *Pomacea canaliculata* | 5.1 | −27.1 |
| *Pomacea canaliculata* | 5.2 | −27.8 |
| *Pomacea canaliculata* | 5.6 | −27.7 |
| *Pomacea canaliculata* | 5.7 | −27.5 |
| *Pomacea canaliculata* | 5.3 | −27.8 |
| *Pomacea canaliculata* | 5.5 | −27.7 |
| *Pomacea canaliculata* | 6.6 | −27.0 |
| *Pomacea canaliculata* | 5.4 | −27.4 |
| *Pomacea canaliculata* | 6.4 | −27.2 |
| *Pomacea canaliculata* | 5.2 | −27.8 |
| *Pomacea canaliculata* | 6.4 | −27.2 |
| *Pomacea canaliculata* | 5.3 | −27.6 |
| *Pomacea canaliculata* | 6.2 | −27.1 |
| *Pomacea canaliculata* | 5.7 | −27.0 |
| *Pomacea canaliculata* | 5.5 | −22.9 |
| *Pomacea canaliculata* | 5.0 | −27.6 |
| *Pomacea canaliculata* | 5.0 | −28.3 |
| *Pomacea canaliculata* | 4.2 | −26.8 |
| *Pomacea canaliculata* | 3.6 | −27.7 |
| *Pomacea canaliculata* | 4.7 | −26.9 |
| *Pomacea canaliculata* | 4.0 | −26.1 |
| *Pomacea canaliculata* | 5.2 | −26.8 |
| *Pomacea canaliculata* | 4.8 | −28.2 |
| *Pomacea canaliculata* | 5.7 | −27.1 |
| *Pomacea canaliculata* | 4.3 | −26.4 |
| *Pomacea canaliculata* | 4.0 | −28.5 |
| *Pomacea canaliculata* | 4.7 | −28.1 |
| *Pomacea canaliculata* | 3.3 | −28.6 |
| *Pomacea canaliculata* | 4.5 | −28.5 |
| *Pomacea canaliculata* | 4.2 | −27.5 |
| *Pomacea canaliculata* | 3.4 | −27.9 |
| *Pomacea canaliculata* | 5.1 | −28.3 |
| *Pomacea canaliculata* | 4.2 | −27.7 |
| *Pomacea canaliculata* | 5.2 | −28.2 |
| *Pomacea canaliculata* | 4.2 | −27.1 |
| *Pomacea canaliculata* | 5.4 | −27.2 |
| *Pomacea canaliculata* | 3.8 | −27.3 |
| *Pomacea canaliculata* | 5.3 | −27.8 |
| *Pomacea canaliculata* | 4.6 | −26.5 |
| *Pomacea canaliculata* | 4.2 | −21.3 |
| *Pomacea canaliculata* | 3.6 | −27.7 |
| *Pomacea canaliculata* | 2.9 | −28.0 |
| *Pomacea canaliculata* | 3.6 | −26.5 |

**Table A2.** Lake Dario recorded individual stable isotope values ($\delta^{15}$N and $\delta^{13}$C) for detritus and apple snails (*Pomacea canaliculata*).

| Sample | $\delta^{15}$N | $\delta^{13}$C |
|---|---|---|
| | (‰, Air) | (‰, VPDB) |
| *Pomacea canaliculata* | 2.5 | −26.2 |
| *Pomacea canaliculata* | 2.5 | −26.0 |
| *Pomacea canaliculata* | 1.9 | −25.1 |
| *Pomacea canaliculata* | 2.9 | −26.4 |
| *Pomacea canaliculata* | 1.8 | −25.9 |
| *Pomacea canaliculata* | 1.4 | −25.3 |
| *Pomacea canaliculata* | 2.3 | −27.9 |
| *Pomacea canaliculata* | 2.8 | −27.0 |
| *Pomacea canaliculata* | 3.4 | −28.4 |
| *Pomacea canaliculata* | 0.3 | −25.9 |
| *Pomacea canaliculata* | 1.5 | −26.0 |
| *Pomacea canaliculata* | 1.0 | −24.4 |
| *Pomacea canaliculata* | 1.2 | −25.9 |
| *Pomacea canaliculata* | 1.0 | −27.0 |
| *Pomacea canaliculata* | 0.8 | −26.3 |
| *Pomacea canaliculata* | 1.5 | −28.7 |
| *Pomacea canaliculata* | 1.5 | −28.2 |
| *Pomacea canaliculata* | 1.8 | −28.5 |
| *Pomacea canaliculata* | 1.9 | −26.0 |
| *Pomacea canaliculata* | 1.1 | −22.3 |
| *Pomacea canaliculata* | 0.3 | −25.9 |
| *Pomacea canaliculata* | −0.1 | −27.1 |
| *Pomacea canaliculata* | 2.3 | −28.9 |
| *Pomacea canaliculata* | 2.1 | −28.2 |
| *Pomacea canaliculata* | 0.5 | −27.2 |
| *Pomacea canaliculata* | 1.9 | −28.5 |
| *Pomacea canaliculata* | 2.0 | −28.9 |
| *Pomacea canaliculata* | 0.4 | −27.1 |
| *Pomacea canaliculata* | 1.7 | −28.4 |
| *Pomacea canaliculata* | 1.9 | −28.9 |
| *Pomacea canaliculata* | 1.5 | −26.9 |
| *Pomacea canaliculata* | 1.5 | −28.1 |
| *Pomacea canaliculata* | 2.3 | −27.6 |
| *Pomacea canaliculata* | 1.1 | −28.3 |
| *Pomacea canaliculata* | 1.9 | −28.5 |
| *Pomacea canaliculata* | 1.9 | −28.1 |
| *Pomacea canaliculata* | 2.2 | −27.4 |
| *Pomacea canaliculata* | 1.5 | −27.4 |
| *Pomacea canaliculata* | 1.7 | −26.7 |
| *Pomacea canaliculata* | 2.5 | −28.8 |
| *Pomacea canaliculata* | 2.6 | −28.4 |
| *Pomacea canaliculata* | 3.2 | −24.5 |
| *Pomacea canaliculata* | 2.4 | −23.6 |
| *Pomacea canaliculata* | 1.3 | −26.2 |
| *Pomacea canaliculata* | 2.3 | −26.9 |
| *Pomacea canaliculata* | 2.9 | −27.6 |
| *Pomacea canaliculata* | 2.9 | −26.6 |
| *Pomacea canaliculata* | 2.1 | −26.6 |
| *Pomacea canaliculata* | 2.9 | −28.2 |
| *Pomacea canaliculata* | 2.7 | −28.5 |
| *Pomacea canaliculata* | 2.2 | −26.4 |
| *Pomacea canaliculata* | 2.6 | −27.1 |
| *Pomacea canaliculata* | 3.6 | −28.3 |
| *Pomacea canaliculata* | 2.4 | −25.9 |
| *Pomacea canaliculata* | 3.4 | −27.5 |
| *Pomacea canaliculata* | 2.6 | −26.6 |
| *Pomacea canaliculata* | 2.4 | −27.5 |
| *Pomacea canaliculata* | 2.3 | −27.6 |
| *Pomacea canaliculata* | 2.4 | −26.7 |
| *Pomacea canaliculata* | 2.9 | −26.8 |
| *Pomacea canaliculata* | 2.7 | −27.1 |
| *Pomacea canaliculata* | 3.0 | −26.1 |
| *Pomacea canaliculata* | 3.1 | −28.3 |
| *Pomacea canaliculata* | 4.2 | −28.6 |

**Table A3.** Chinese lake site recorded individual stable isotope values ($\delta^{15}$N and $\delta^{13}$C) for detritus and apple snails (*Pomacea canaliculata*).

| Sample | $\delta^{15}$N | $\delta^{13}$C |
|---|---|---|
| | (‰, Air) | (‰, VPDB) |
| *Pomacea canaliculata* | 3.2 | −28.9 |
| *Pomacea canaliculata* | 2.9 | −28.0 |
| *Pomacea canaliculata* | 3.6 | −27.6 |
| *Pomacea canaliculata* | 3.5 | −28.7 |
| *Pomacea canaliculata* | 4.1 | −25.0 |
| *Pomacea canaliculata* | 3.0 | −29.1 |
| *Pomacea canaliculata* | 2.8 | −29.5 |
| *Pomacea canaliculata* | 2.8 | −28.0 |
| *Pomacea canaliculata* | 4.1 | −29.3 |
| *Pomacea canaliculata* | 3.5 | −28.2 |
| *Pomacea canaliculata* | 3.8 | −28.7 |
| *Pomacea canaliculata* | 1.8 | −27.7 |
| *Pomacea canaliculata* | 3.4 | −28.4 |
| *Pomacea canaliculata* | 2.3 | −28.5 |
| *Pomacea canaliculata* | 3.4 | −28.4 |
| *Pomacea canaliculata* | 2.6 | −27.6 |
| *Pomacea canaliculata* | 3.1 | −28.3 |
| *Pomacea canaliculata* | 3.2 | −29.0 |
| *Pomacea canaliculata* | 3.5 | −26.8 |
| *Pomacea canaliculata* | 2.7 | −28.6 |
| *Pomacea canaliculata* | 2.9 | −27.5 |
| *Pomacea canaliculata* | 2.9 | −29.4 |
| *Pomacea canaliculata* | 3.3 | −26.8 |
| *Pomacea canaliculata* | 2.5 | −28.2 |
| *Pomacea canaliculata* | 2.7 | −28.1 |
| *Pomacea canaliculata* | 5.7 | −28.5 |
| *Pomacea canaliculata* | 4.7 | −28.2 |
| *Pomacea canaliculata* | 3.9 | −26.7 |
| *Pomacea canaliculata* | 4.5 | −28.6 |
| *Pomacea canaliculata* | 5.7 | −27.8 |
| *Pomacea canaliculata* | 4.7 | −27.7 |
| *Pomacea canaliculata* | 5.1 | −28.6 |
| *Pomacea canaliculata* | 5.4 | −27.7 |
| *Pomacea canaliculata* | 5.3 | −28.6 |
| *Pomacea canaliculata* | 4.4 | −27.8 |
| *Pomacea canaliculata* | 5.2 | −28.8 |
| *Pomacea canaliculata* | 4.8 | −28.2 |
| *Pomacea canaliculata* | 5.0 | −28.7 |
| *Pomacea canaliculata* | 4.5 | −27.9 |
| *Pomacea canaliculata* | 4.8 | −28.1 |
| *Pomacea canaliculata* | 4.6 | −27.6 |
| *Pomacea canaliculata* | 5.1 | −27.9 |
| *Pomacea canaliculata* | 4.5 | −27.9 |
| *Pomacea canaliculata* | 4.8 | −28.4 |
| *Pomacea canaliculata* | 5.5 | −28.1 |
| *Pomacea canaliculata* | 5.1 | −27.9 |
| *Pomacea canaliculata* | 5.2 | −28.7 |
| *Pomacea canaliculata* | 4.5 | −28.2 |
| *Pomacea canaliculata* | 4.1 | −28.4 |
| *Pomacea canaliculata* | 4.8 | −28.5 |

**Table A4.** Chinese creek site recorded individual stable isotope values ($\delta^{15}$N and $\delta^{13}$C) for detritus and apple snails (*Pomacea canaliculata*).

| Sample | $\delta^{15}$N | $\delta^{13}$C |
|---|---|---|
| | (‰, Air) | (‰, VPDB) |
| *Pomacea canaliculata* | 3.4 | −30.5 |
| *Pomacea canaliculata* | 5.9 | −30.9 |
| *Pomacea canaliculata* | 5.5 | −30.8 |
| *Pomacea canaliculata* | 5.2 | −30.7 |
| *Pomacea canaliculata* | 2.7 | −31.3 |
| *Pomacea canaliculata* | 6.6 | −30.3 |
| *Pomacea canaliculata* | 7.4 | −28.6 |
| *Pomacea canaliculata* | 3.7 | −31.2 |
| *Pomacea canaliculata* | 9.4 | −28.9 |
| *Pomacea canaliculata* | 4.8 | −30.2 |
| *Pomacea canaliculata* | 8.3 | −29.3 |
| *Pomacea canaliculata* | 7.1 | −29.8 |
| *Pomacea canaliculata* | 7.1 | −30.5 |
| *Pomacea canaliculata* | 3.7 | −30.9 |
| *Pomacea canaliculata* | 9.1 | −28.5 |
| *Pomacea canaliculata* | 9.8 | −28.4 |
| *Pomacea canaliculata* | 6.9 | −30.1 |
| *Pomacea canaliculata* | 6.0 | −30.1 |

**Table A5.** Kawainui Marsh site recorded individual stable isotope values ($\delta^{15}$N and $\delta^{13}$C) for detritus and apple snails (*Pomacea canaliculata*).

| Sample | $\delta^{15}$N | $\delta^{13}$C |
|---|---|---|
| | (‰, Air) | (‰, VPDB) |
| *Pomacea canaliculata* | 8.5 | −18.0 |
| *Pomacea canaliculata* | 7.3 | −19.4 |
| *Pomacea canaliculata* | 7.5 | −19.8 |
| *Pomacea canaliculata* | 8.8 | −25.0 |
| *Pomacea canaliculata* | 5.5 | −24.9 |
| *Pomacea canaliculata* | 8.0 | −22.1 |
| *Pomacea canaliculata* | 9.8 | −18.2 |
| *Pomacea canaliculata* | 10.2 | −26.6 |
| *Pomacea canaliculata* | 9.5 | −23.9 |
| *Pomacea canaliculata* | 7.2 | −19.6 |
| *Pomacea canaliculata* | 8.3 | −20.1 |
| *Pomacea canaliculata* | 8.5 | −20.3 |
| *Pomacea canaliculata* | 7.5 | −16.5 |
| *Pomacea canaliculata* | 7.4 | −18.7 |
| *Pomacea canaliculata* | 7.5 | −22.9 |
| *Pomacea canaliculata* | 7.9 | −24.7 |
| *Pomacea canaliculata* | 8.5 | −25.7 |

*Appendix A.5.*

Tables A6–A10: Lists of species identities, the means by which identifications were made, and the mean and standard deviation for corresponding $\delta^{15}$N and $\delta^{13}$C values recorded for all food resources (including plants, plankton, detritus, phytoplankton, zooplankton, and seston) used in SIAR models for all collection sites.

**Table A6.** Species identities, means of identification, and mean and standard deviation for all food resources (plants, plankton, detritus, and phytoplankton) used in SIAR models for Lake Sauce in Maldonado, Uruguay.

| Plants /Plankton Detritus /Phytoplankton | Sample Size (n) | Species ID | Means of ID | $\delta^{15}$N | | $\delta^{13}$C | |
|---|---|---|---|---|---|---|---|
| | | | | (‰, Air) | $\delta^{15}$N Std Dev. | (‰, VPDB) | $\delta^{13}$C Std Dev. |
| Aquatic Plant 1 | 5 | *Certophyllum* sp. | Sequence Data | 6.6 | 0.6 | −26.6 | 1.2 |
| Aquatic Plant 2 | 10 | *Salvinia* sp. | Sequence Data | 5.0 | 0.5 | −29.2 | 0.8 |
| Aquatic Plant 3 | 10 | *Egeria* sp. | Sequence Data | 5.9 | 1.1 | −26.7 | 0.8 |
| Aquatic Plant 4 | 10 | **Monocot Perennial** | Morphological Characteristics | 4.9 | 1.0 | −28.3 | 0.7 |
| Aquatic Plant 5 | 10 | *Ludwigia* sp. | Sequence Data | 5.2 | 0.8 | −28.2 | 0.5 |
| Aquatic Plant 6 | 10 | *Eichhornia crassipes* | Sequence Data | 5.5 | 0.3 | −27.9 | 0.3 |
| Aquatic Plant 7 | 10 | *Echinodorus* sp. | Sequence Data | 4.7 | 0.9 | −27.4 | 0.9 |
| Terrestrial Plant 1 | 10 | *Poaceae* sp. | Sequence Data | 5.6 | 2.4 | −12.4 | 0.9 |
| Terrestrial Plant 2 | 5 | *Persicaria* sp. | Sequence Data | 2.6 | 0.9 | −30.8 | 1.0 |
| Terrestrial Plant 3 | 10 | *Breynia* sp. | Sequence Data | 4.3 | 1.5 | −29.7 | 0.8 |
| Terrestrial Plant 4 | 10 | *Hydrocotyle* sp. | Sequence Data | 3.1 | 0.6 | −30.7 | 0.4 |
| Terrestrial Plant 5 | 10 | *Rumex* sp. | Sequence Data | 4.2 | 1.4 | −30.3 | 0.4 |
| Terrestrial Plant 6 | 5 | **Not Identified** | **N/A** | 5.5 | 1.2 | −27.4 | 0.5 |
| Zooplankton | 11 | **N/A** | **N/A** | 5.0 | 1.5 | −29.1 | 1.9 |
| Detritus | 5 | **N/A** | **N/A** | 2.83 | 0.8 | −26.5 | 0.7 |
| Phytoplankton | 3 | **N/A** | **N/A** | 4.7 | 0.9 | −28.3 | 0.4 |

**Table A7.** Species identities, means of identification, and mean and standard deviation for all food resources (plants, detritus, zooplankton, and seston) used in SIAR models for Lake Dario in Maldonado, Uruguay.

| Plants/ Detritus Zooplankton/ Seston | Sample Size (n) | Species ID | Means of ID | $\delta^{15}$N | | $\delta^{13}$C | |
|---|---|---|---|---|---|---|---|
| | | | | (‰, Air) | $\delta^{15}$N Std Dev. | (‰, VPDB) | $\delta^{13}$C Std Dev. |
| Aquatic Plant 1 | 10 | *Salvinia* sp. | Sequence Data | 2.7 | 0.6 | −28.5 | 0.3 |
| Aquatic Plant 2 | 10 | *Ludwigia* sp. | Sequence Data | 3.6 | 2.3 | −27.7 | 0.6 |
| Aquatic Plant 3 | 10 | *Egeria densa* | Sequence Data | 3 | 0.6 | −29.4 | 1.3 |
| Aquatic Plant 4 | 10 | *Potamogeton* sp. | Sequence Data | −0.04 | 0.8 | −23.9 | 1.4 |
| Aquatic Plant 5 | 10 | *Pistia stratiotes* | Sequence Data | 4.3 | 0.5 | −29 | 0.5 |
| Aquatic Plant 6 | 10 | *Potamogeton* sp. | Sequence Data | 3 | 1.8 | −23.9 | 1.0 |
| Aquatic Plant 7 | 10 | *Eichhornia crassipes* | Sequence Data | 2.3 | 0.6 | −27.8 | 0.9 |
| Aquatic Plant 8 | 2 | *Spirogyra* sp. (Alga) | Microscopy | 0.9 | 0.02 | −21.0 | 0.04 |
| Terrestrial Plant 1 | 5 | *Poaceae* sp. | Sequence Data | 7.9 | 0.36 | −13.2 | 0.1 |
| Terrestrial Plant 2 | 10 | *Ludwigia* sp. | Sequence Data | 6.0 | 1 | −29.5 | 0.7 |
| Detritus | 5 | **N/A** | **N/A** | 1.3 | 0.8 | −28.0 | 0.7 |
| Zooplankton | 3 | **N/A** | **N/A** | 3.4 | 0.02 | −25.4 | 1.4 |
| Seston | 3 | **N/A** | **N/A** | 1.7 | 0.2 | −30.5 | 0.1 |

**Table A8.** Species identities, means of identification, and mean and standard deviation for all food resources (plants, detritus, and zooplankton) used in SIAR models for the Chinese lake site.

| Plants/ Detritus | Sample Size (n) | Species ID | Means of ID | $\delta^{15}$N (‰, Air) | $\delta^{15}$N Std Dev. | $\delta^{13}$C (‰, VPDB) | $\delta^{13}$C Std Dev. |
|---|---|---|---|---|---|---|---|
| Aquatic Plant 1 | 10 | *Thalia* sp. | Sequence Data | 5.0 | 0.8 | −30.6 | 0.9 |
| Aquatic Plant 2 | 10 | *Zizania* sp. | Sequence Data | 3.5 | 2.1 | −28.5 | 0.4 |
| Aquatic Plant 3 | 10 | *Aternanthera* sp. | Sequence Data | 5.6 | 0.2 | −28.5 | 0.7 |
| Aquatic Plant 4 | 5 | *Carex* sp. | Sequence Data | 3.3 | 1.0 | −31.5 | 0.7 |
| Terrestrial Plant 1 | 5 | *Glechoma* sp. | Sequence Data | −3.3 | 0.4 | −37.4 | 1.5 |
| Terrestrial Plant 2 | 5 | *Oplismenus* sp. | Sequence Data | 1.3 | 0.6 | −34.6 | 0.7 |
| Terrestrial Plant 3 | 5 | *Erigeron* sp. | Sequence Data | 0.3 | 0.5 | −33.3 | 0.7 |
| Terrestrial Plant 4 | 5 | **Not identified** | **N/A** | −0.5 | 0.6 | −31.1 | 0.6 |
| Terrestrial Plant 5 | 5 | *Nastus* sp. | Sequence Data | 1.4 | 0.3 | −30.6 | 0.2 |
| Detritus | 5 | **N/A** | **N/A** | 2.9 | 1.0 | −29.2 | 1.4 |
| Zooplankton | 4 | **N/A** | **N/A** | 3.3 | 0.4 | −26.8 | 0.2 |

**Table A9.** Species identities, means of identification, and mean and standard deviation for all food resources (plants and detritus) used in SIAR models for the Chinese creek site.

| Plants/ Detritus | Sample Size (n) | Species | Means of ID | $\delta^{15}$N (‰, Air) | $\delta^{15}$N Std Dev. | $\delta^{13}$C (‰, VPDB) | $\delta^{13}$C Std Dev. |
|---|---|---|---|---|---|---|---|
| Aquatic Plant 1 | 10 | *Oenathe* sp. | Sequence Data | 5.3 | 1.6 | −30.8 | 1.3 |
| Aquatic Plant 2 | 10 | *Hygrophila* sp. | Sequence Data | 4.8 | 1.8 | −31.5 | 1.0 |
| Aquatic Plant 3 | 10 | *Iris pseudacarus* | Sequence Data | 6.7 | 1.4 | −31.1 | 0.9 |
| Terrestrial Plant 1 | 12 | *Pteris henryi* | Sequence Data | 1.3 | 2.0 | −29.0 | 1.3 |
| Terrestrial Plant 2 | 10 | *Oplismenus* sp. | Sequence Data | −4.2 | 4.1 | −31.3 | 0.7 |
| Detritus | 5 | **N/A** | **N/A** | −0.1 | 3.3 | −21.0 | 2.4 |

**Table A10.** Species identities, means of identification, and mean and standard deviation for all food resources (plants and detritus) used in SIAR models for Kawainui Marsh.

| Plants/ Detritus | Sample Size (n) | Species ID | Means of ID | $\delta^{15}$N (‰, Air) | $\delta^{15}$N Std Dev. | $\delta^{13}$C (‰, VPDB) | $\delta^{13}$C Std Dev. |
|---|---|---|---|---|---|---|---|
| Aquatic Plant 1 | 5 | *Bacopa monnieri* | Field ID Manual | 9.9 | 1.4 | −30.1 | 0.3 |
| Aquatic Plant 2 | 5 | *Brahchiaria mutica* | Field ID Manual | 7.5 | 0.3 | −12.4 | 0.3 |
| Aquatic Plant 3 | 5 | *Echinochloa crusgalli* | Field ID Manual | 8.9 | 1.0 | −12.3 | 0.3 |
| Terrestrial Plant 1 | 5 | *Cynodon dactylon* | Field ID Manual | 9.0 | 0.6 | −12.6 | 0.5 |
| Detritus | 5 | **N/A** | **N/A** | 3.4 | 1.4 | −16.2 | 1.6 |

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
