# Peer review of "Invasive Apple Snail Diets in Native vs. Non-Native Habitats Defined by SIAR (Stable Isotope Analysis in R)"

_sustainability, doi:10.3390/su14127108_

Round 1

Reviewer 1 Report

Throughout the text: I suggest to replace "Pomacea" with "P." after its first metion in the paragraph. In addition, you should either use latin name or common name of studied taxon instead of using both separately and tugether.

L61-90: In this part I after short explanation of isotope role in food web study I was expecting to see more detailed hypothesis presented instead of decription of sampling sites. This later paragraph largerly belongs to respective methods section where site description is short. I suggest here to develop hypothesis and the expectation and move site descriptions to method section. 

SIAR abbreviation is numerously explained in the intro and methods section. Perhaps single explanation is enough.

Methods section: the genetic methodology is too lengthy, especially unnecessary details of DNA extraction procedure and gel electrophoresis are given. I suggest to shorten gentic methdology part significantly.

Results section: A lot of text is written about the genetics which seems to be only used for plant species identification. Why you needed barcoding of zooplankton if you did not use taxa level information? 

In addition there is no any statistical output in the manuscrip. Was there any significant changes ins stable isotop composition between lakes? any other statistical details? 

L309: delete repeated "of the diet"

L388: "3.6. Figures and 3.7. Tables". What the purpose of these subchapters ? Why you did not just attach each figure and table to the respective subchapter before (i.e. each figure and table within the description of respective lake results).

Tables: "AP acronym refers to aquatic plant species and TP acronym refers to terrestrial plant species." change to: "AP - aquatic plant; TP - terrestrial plant"

Tables: Does the last collumn undicate the source of identification? If so, maybe idicate that in more precisely.

L430-431 and L472: I do not understand what this means. species utilizes more diet in one lake compared to other? What it means? Are they starving? Stressed? Resources are limmited? Need to be better explained. Although next paragraph says that plant species are on lake Dario less abundant, however apple snail shift its diet to other source - so where is the sign of limmited food utiization? 

Similar to intro section, the discussion section lacks the clear answer on the hypotheses stated. This because no hypotheses were developed fully and no expectations were drawn. At the end I suggest to state clear hypotheses and then discuss the obtained results under the expectations of that hypitheses. 

Author Response

Thank you for your time and consideration of my manuscript. I have tried to address you concerns point-by-point in the affixed Word file.

Scriber

Reviewer 2 Report

The article investigates why invasive apple snails (Pomacea canaliculata) differentially impact native and non-native habitats, testing the hypothesis that apple snails (Pomacea canaliculata) shift their trophic ecology and/or feeding behavior between native and non-native habitats.

I consider the results very interesting, where it was pointed out the ecological imbalance that the anthropological impacts can cause in the diet of Pomacea canaliculata and in the intra and interspecific relationships of the species. The organism ceases to be a generalist and becomes a specialist, where there may be greater overlapping of niches and possibly a competitive exclusion of native species by invasive species.  

General considerations:

Consider inserting more up-to-date references in the article.

Line 106: How many  apple snail  were collected? Were all the same size and age collected? In this case, wouldn't it be necessary to standardize the age and size of the organisms, since the diet can change depending on these parameters?

Line 106: How many plants were collected? Which plants (species) were analyzed? What tissues were used for the analyses?

Line 106: Mention the methodology used to collect and preserve the analyzed tissues.

Line 107: What is the condition of tissue preservation? At what temperature were they preserved and for how long?

Line 108: What tissue was removed for analysis?

Line 115: Improve the definition of what this debris would be. Ground? Sediment?

Line 115: How deep is the detritus collection? How was such depth defined?

Line 121: How long were the samples preserved? Which standard indicates the allowed period of preservation?

Line 123: Mention the methodology corresponding to plankton collection.

Line 123: What criteria did you use to select the type of filter? In this case 50 µm (Zooplankton) and 20 µm (phytoplankton). Make this clear in the text.

Line 129: When you say separated by hand, exemplify the method. How would it be? Using microscopy, tweezers?

Line 297: Reduce the amount of information inside the parentheses of Figure 1. The text becomes a bit long. Try to insert the information in a natural way within the text, so that the reading becomes more fluid.

Do the same for the lines 320 (Figure 2), 357 (Figure 3), 378 (Figure 4) and 488 (Figure 5).

Line 488: Detail better what would be the ecological impacts described in this section, on non-native habitats. Insert references.

Author Response

Thank your for your time and consideration of my manuscript. Please see I have tried to address your questions and concerns. I have affixed Word file with a point0by-pooint reply to your comments.

Scriber

Round 2

Reviewer 1 Report

I apriciate the autor responses to my comments and do not see additional room for the ms improvement at this stage. I suugest to acept the manuscript in its cuurent form.